# Smokers' Attitude and Behavior towards Cigarette Littering in Romania: A Survey-Based Approach

**Elena Simina Lakatos** [1,2]**, Lucian Ionel Cioca** [3,]*****, Andrea Szilagyi** [1,2]**, Andreea Loredana Bîrgovan** [1,2]
**and Elena Cristina Rada** [4]

[1] Institute for Research in Circular Economy and Environment Ernest Lupan, 400800 Cluj-Napoca, Romania; simina.lakatos@ircem.ro (E.S.L.); andrea.szilagyi@ircem.ro (A.S.); loredana.birgovan@ircem.ro (A.L.B.)

[2] Faculty of Industrial Engineering, Robotics and Product Management, Technical University of Cluj-Napoca, 400641 Cluj-Napoca, Romania

[3] Faculty of Engineering, Lucian Blaga University of Sibiu, 10 Victoriei Blv., 550024 Sibiu, Romania

[4] Theoretical and Applied Science Department, Insubria University, 46 Via G.B. Vico, I-21100 Varese, Italy; elena.rada@uninsubria.it

***** Correspondence: lucian.cioca@ulbsibiu.ro

**Abstract:** Cigarette butts continue to be a significantly detrimental challenge for both human health and the quality of the environment and life in general. The escalating accumulation of inadequately discarded cigarette butts continues unabated, in spite of the myriad legislative procedures that have been instituted by authorities with the objective of incentivizing diminution of this phenomenon. For decades, the scientific literature has discussed the importance of smokers' behavior and beliefs in contributing to the problem. Thus, the objective of this study is to analyze cigarette butt littering behavior using a survey-based questionnaire. A total of 1643 complete responses were collected from Romanian smokers addressing their knowledge, practices, and views regarding the disposal of cigarette butts. According to the findings, awareness about the impacts and characteristics of cigarette butts is problematic, as are smokers' self-reported explanations for their conduct. Specifically, more than 30% of the sample stated that cigarette butts are biodegradable, and 31.5% believe that cigarette butts are not toxic to the environment. The results also showed that only 19.7% of the smokers strongly believed that they should be considered accountable for their behavior. Future study directions are provided to advance studies in this area and improve present approaches to lessen the frequency of this behavior.

**Keywords:** pollution; microplastic; cigarette waste; littering behavior

## 1. Introduction

Although there are many different sorts of pollution, one of the more noticeable types is littering [1]. A common definition of litter is "waste, discarded or scattered about in disorder across a socially unacceptable area", while there are other definitions that can be used as well [2,3]. Spreading waste is referred to as "littering", and it can be either active or passive. Active littering is a deliberate action that refers to the litter retained (in hands, pockets, bags, or any other personal items) and purposely placed in the area of living before departing [4]. The definition of passive littering, on the other hand, is "litter [that] is placed in an area that is utilized; after leaving the area, waste is left behind" [4,5].

Due to its hazardous externalities on the natural environment and public health, this behavior deserves the attention of the public. The environment may be severely damaged by items like cigarettes, glass, plastic, take-out food packaging, bottles, and many others. There are extra environmental costs to consider besides the costs of hiring someone to collect the litter because some of these items are not biodegradable. In the wild, littering

may seriously harm the ecosystem, since it is not only unappealing, but also very dangerous, as a glass bottle or a thrown cigarette could easily start a deadly forest fire [6,7].

Cigarette waste (particularly cigarette butts and the filters of heated tobacco products) is one of the most prevalent types of litter found in outdoor spaces. The impact it has on urban environments cannot be overstated. Cigarette butts account for the largest share of waste that is found improperly discarded on surfaces during clean-up processes around the world [8]. The fact that this type of waste is non-biodegradable and extremely harmful to marine flora and fauna as it exhibits a unique mixture of physical and chemical contamination are additional disadvantages. The incorporation of cigarette butts into recycling schemes and strategies has been the subject of numerous projects and initiatives; however, currently, collection strategies are inadequate due to the current system's inefficiency at the municipal level and citizens' poor environmental behavior. Also, as environments are becoming more complex and require advanced analysis techniques to control the environmental causalities attributed to smoking, the scientific literature has begun studying advanced techniques such as Machine Learning or Artificial Intelligence in order to better understand cigarette consumption and littering [9,10].

As consumer behavior is believed to be one of the explanatory factors (besides poor infrastructure and lack of awareness of the negative impacts) of cigarette butt littering, this study aims to explore cigarette littering in Romania through the following factors at the consumer level: (1) smokers' motivation for littering; (2) smoking and littering behavior; (3) smokers' accountability perceptions, and (4) environmental knowledge about cigarette waste. The authors designed a questionnaire based on the existing literature, which was then validated by domain experts to gain an understanding of all of these variables. The questionnaire was completed by 1643 respondents from all Romanian regions.

### 1.1. Environmental Impact of Improper Cigarette Butt Disposal

Pollution affects more than 200 million people worldwide, with littering playing a significant role [11]. Despite major initiatives and campaigns to combat the inappropriate disposal of waste, it remains a significant problem in many parts of the world. The problem of waste contamination is widespread, affecting both developed and developing countries.

Cigarettes pose a risk even when they are not smoked, since the components decompose significantly more quickly when they come into contact with water [12]. While some compounds can disintegrate more quickly and be absorbed by the environment, others may persist for a longer time. According to estimates by Farzadkia et al. [13], nickel, lead, and zinc are released into the environment starting on the first day of exposure. The entire process of breaking down these filters can take more than ten years, which is a negative step toward microplastic contamination. About 15,000 microplastic fibers are contained in each cigarette filter, and 100 of those fibers can be shed into the water every day [14].

It is, therefore, not unexpected that a lot of research has been conducted to better understand the disposal behavior of cigarette butts considering that they are the most disposed of type of waste globally. However, most have offered little insight and produced less obvious pollution reductions [15]. The causes of this practice include habits, convenience, a lack of ashtrays and trash cans, inadvertent disposal of the cigarette due to its tiny size, and the misconception that cigarette butts naturally decompose [4,16].

Considering the difficulty in collecting cigarette butts after they have been released into the environment, exact estimates of the quantity of cigarette butts at the national level are scarce, and more limited still at the local or regional level. There is currently no mention of the amount of waste produced or any countermeasures in the National Waste Management Plan of Romania. According to an estimate by Tobacco Atlas based on data from the national level and the methodology proposed by Novotny and Zhao [16], there are approximately 5924 tons of waste produced by cigarette packaging each year in Romania, 3357 tons of waste produced by cigarettes butts, and a total of 9281 tons of waste related to packaging and cigarette butts. Given that an average cigarette weighs 0.2 g and that the

Tobacco Atlas recorded 3.357 tons of tobacco consumption, this equates to 16,785,000,000 cigarettes being tossed into the environment annually in Romania.

### 1.2. Behavioral Aspects of Cigarette Littering

Humans have a direct impact on the environment through passive conduct and negligence in waste management, as well as a lack of responsibility or concern for proper disposal. The entire transformation must begin with the users. It is critical to gain a better understanding of how environmental attitudes and perceived efficacy influence citizens' behavior in relation to the issue of cigarette waste generation, and whether there are significant disparities in the potential environmental and social repercussions associated with this waste between knowledgeable and uninformed individuals. With respect to general littering behavior, according to Ojedokun [1], low self-efficacy and low locus of control lead to a positive attitude toward littering, while better education and awareness lead to a negative attitude toward littering. Marital status, monthly earnings, religious convictions, level of education, age, and type of living all have an impact on littering behavior, according to Arafat et al. [17] and Al-Khatib et al. [18]. Littering is also more common when a person is in a hurry, the item is biodegradable, there is an expectation that someone else will pick it up, and the item is not recyclable. Several studies also claim that littering is frequently the result of a lack of a waste container nearby.

Existing literature on cigarette waste littering specifically indicates that a variety of factors may influence cigarette butt littering behavior [19]. For example, Rath et al. [15] discovered that smokers' beliefs about cigarette butts not being litter only predicted improper discarding behavior among smokers aged 18 and above. They also discovered that males throw significantly more cigarette butts on the ground than females. According to Schultz et al. [19], distance to waste bin significantly predicted improper cigarette butt disposal in outdoor public places in the United States. In addition, Miller and Burbach [20] claimed that environmental views and awareness, as well as habits, had an impact on smokers who disposed of their cigarette butts on Jekyll Island.

### 1.3. Legislative Actions Aimed at Reducing Cigarette Butt Littering

The European Union (EU) Directive on reducing the environmental impact of certain plastic products, also known as the Single-Use Plastics Directive [21] (Directive 2019/904), aims at preventing and reducing the environmental impact of certain plastic products while also promoting the shift to a circular economy. This includes several actions targeted to the directive's items, such as EU-level restrictions on single-use plastic products when other options are available. This method went into effect on 3 July 2019, with the deadline for domestic adaptation of the directive extended until 3 July 2021, and time set aside for the proposal and approval of relevant measures for the directive's successful implementation in member nations.

This European legislative framework covers single-use plastics, and the directive includes tobacco product filters and filters marketed for use in conjunction with tobacco products containing plastic. Thus, it is vital to lessen the significant environmental impact caused by waste created at the end of the use of tobacco products with filters that contain plastic and are thrown directly into the environment rather than in a properly arranged location. Extended liability plans for producers placing cigarette filters using single-use plastics on the European market will provide viable, sustainable alternatives, reducing the negative environmental impact [21,22].

In the case of filters for tobacco products, separate collection to ensure proper waste treatment according to the waste hierarchy is not mandatory..

## 2. Materials and Methods

The design of this study is cross-sectional based on the survey approach. This means that the variables of interest in the study were surveyed only once, in a single sample,

using a questionnaire. The snowballing technique was used for sample selection, and the online survey created on the e-survey platform was distributed online across all regions of Romania. The mandatory characteristics of the respondents in the target group of the study were (1) adults exclusively over 19 years old, (2) active smokers, and (3) residents of Romania. The authors created the questionnaire on the online platform and responded to any issues or inquiries participants had about their participation in the research.

A total of 2046 questionnaires were completed in January 2022. Of the 2046 responses received, 199 incomplete responses or those not fulfilling the mandatory inclusion criteria were identified after database analysis and were removed from the study so as not to affect the validity of the measurement procedure. After applying the mandatory exclusion criteria, the valid sample was composed of 1643 responses. Table 1 illustrates the socio-demographic characteristics of the participants in detail. All study participants agreed to participate and gave permission for their data to be used for research purposes. The participants received notice that their information would remain strictly confidential, and data collection was fully voluntary and anonymous.

**Table 1.** Socio-demographic characteristics of the sample.

|  | **Criteria** | **%** |
|---|---|---|
| Gender | Female | 48.4% |
|  | Male | 51.6% |
| Age | 20–30 years | 7% |
|  | 31–40 years | 37.2% |
|  | 41–50 years | 38.8% |
|  | 51–60 years | 16.9% |
|  | 61–70 years | 0.1% |
| Occupational status | Employee | 57.2% |
|  | Entrepreneur | 19.7% |
|  | Student | 0.8% |
|  | Unemployed | 11.4% |
|  | Housewife/men | 10.7% |
|  | Other | 0.2% |
|  | Primary school | 18.9% |
| Education | Middle school | 18.5% |
|  | High school | 19.4% |
|  | Bachelor's degree | 22.3% |
|  | Master's degree | 20.6 |
|  | Doctoral studies | 0.3% |
| Residence | Urban | 81% |
|  | Rural | 19% |

## 3. Results

### 3.1. Smoking Behavior

Firstly, a multiple-choice question was presented to identify the type of cigarette products consumed by our sample. A well-defined majority of respondents (98.3%) stated that they smoke classic cigarettes daily. Heated tobacco products are the less prevalent option: only 0.9% of the total sample stated that they smoke daily, 49.2% rarely, and 49.9% never.

Secondly, the data in Table 2 presents the self-reported number of cigarettes smoked by the respondents on a typical day. As can be easily observed, there is a significant difference between the number of cigarettes smoked per day at home and those smoked in public spaces. Since locations outside the home are useful in identifying the pattern of

consumption and involvement in cigarette throwing, the table shows the number of cigarettes smoked in public spaces in detail. The highest number of cigarettes consumed is reported at the workplace, in specially arranged places. Percentages were obtained by referencing the mean of the cigarette consumption recorded at the sample level.

**Table 2.** Reported smoking locations.

| | Location | Percentage of Consumed Cigarettes | Maximum Number of Consumed Cigarettes |
|---|---|---|---|
| | Inside the personal domicile | 3.65% | 30 |
| | At work | 51.7% | 18 |
| Cigarettes consumed in public places | Subway exit | 0.4% | 4 |
| | While waiting at traffic lights | 1.28% | 5 |
| | In the park | 2.14% | 5 |
| | On the street | 2.23% | 5 |
| | Public transport stations | 2% | 2 |
| | Parking lots | 8.11% | 5 |
| | In front of the block | 0.85% | 4 |
| | In front of public institutions | 0.85% | 3 |
| | Other public places | 2.57% | 10 |

*3.2. Littering Motivations*

Two questions with 'no' and 'yes' answer selections were administered to the respondents to determine whether they throw cigarette butts on the ground while walking or directly from the car. The percentages obtained indicated that 17% of the entire sample throw cigarette butts on the ground, and 8.4% of the respondents admitted to throwing a cigarette butt from the car directly onto the footpath at least once. The table below (Table 3) shows the detailed situation by age group along with the self-reported reasons why the participants proceeded to improperly dispose of the litter. It is also important to mention that some motivations appear to increase in frequency depending on the age group. For example, only 3.3% of 20–31 years old reported that they litter cigarette butts because it is the right thing to do, whilst 18.3% of the respondents aged between 40 and 51 years old reported the same reason. The same pattern of increased frequency can be observed with the self-reported reasons regarding the absence of a penalty ('no one fined me') and the belief that littering is acceptable if you are not warned by someone when you are littering cigarette butts.

**Table 3.** Self-reported motivations for cigarette littering.

| Age | Percentage of Respondents Disposing Cigarettes Improperly | Self-Reported Reasons for Cigarette Littering | |
|---|---|---|---|
| 20–31 years old | 26.1% | Because I believe it's the right thing to do | 3.3% |
| | | Because I was in a hurry | 13.3% |
| | | Because I couldn't find a trash can or ash-tray nearby | 63.3% |
| | | Because no one fined me | 6.7% |
| | | Because no one told me it was wrong | 6.7% |
| | | Because other smokers throw them on the ground too | 6.7% |
| 30–41 years old | 10% | Because I believe it's the right thing to do | 23.0% |
| | | Because I was in a hurry | 19.7% |
| | | Because I couldn't find a trash can or ash-tray nearby | 14.8% |
| | | Because no one fined me | 21.3% |
| | | Because no one told me it was wrong | 21.3% |
| | | Because other smokers throw them on the ground too | 23.0% |
| 40–51 years old | 12.9% | Because I believe it's the right thing to do | 18.3% |
| | | Because I was in a hurry | 20.7% |
| | | Because I couldn't find a trash can or ash-tray nearby | 23.2% |
| | | Because no one fined me | 18.3% |
| | | Because no one told me it was wrong | 18.3% |
| | | Because other smokers throw them on the ground too | 18.3% |
| 51–60 years old | 14.1% | Because I believe it's the right thing to do | 17.9% |
| | | Because I was in a hurry | 17.9% |
| | | Because I couldn't find a trash can or ash-tray nearby | 7.7% |
| | | Because no one fined me | 28.2% |
| | | Because no one told me it was wrong | 28.2% |
| | | Because other smokers throw them on the ground too | 17.9% |

*3.3. Environmental Knowledge about Cigarette Waste Pollution*

In terms of the understanding that the surveyed smokers have concerning the impacts of cigarette butts on the environment when they are incorrectly discarded, the findings appear to be consistent with what has already been discovered in the literature. For example, more than 30% of smokers stated that cigarette butts are biodegradable, and 31.5% believe that cigarette butts are not toxic to the environment. Moreover, it is also important to note that in this sample, the smokers appear to acknowledge the economic impact of cigarette littering, as 35.9% of the respondents claimed that is true that cigarette littering represents an issue with economic repercussions (Table 4).

**Table 4.** Environmental knowledge on cigarette waste effects on environment.

| Statement | True | False | I Do Not Know |
|---|---|---|---|
| Cigarette butts are biodegradable | 31.3% | 35.0% | 33.7% |
| Cigarette butts are toxic to the environment | 35.8% | 31.5% | 32.7% |
| Cigarette butts are dangerous to animals | 37.8% | 30.1% | 32.1% |
| Cigarette butts are dangerous if they are thrown in the trash | 34.3% | 34.2% | 31.5% |
| Cigarette butts are dangerous to aquatic ecosystems | 37.9% | 32.2% | 29.9% |
| Cigarette littering is an issue with economic impact | 35.9% | 32.3% | 31.8% |
| Cigarette littering is a problem with social impact | 34.0% | 31.0% | 35.0% |

*3.4. Perceived Accountability of Cigarette Littering*

Concerning the assigned responsibility for the cigarette butt littering phenomenon from the standpoint of customers, it appears to be quite diffusely distributed in this sample, with no significant percentage differences across the four responsible categories: smokers, cigarette producers, local administrations, and the central government. Only 19.7% of the smokers surveyed strongly believed that they should be considered responsible for this phenomenon, while a similar percent of 18.9% considered that the central government is to blame for not taking enough measures to discourage such behaviors. Therefore, this diffuse pattern of perceived responsibility might imply that future interventions aimed at reducing cigarette littering should also take into account the involvement of multiple actors involved directly or indirectly (smokers, tobacco product manufacturers, local and central governments, etc.) in the cigarette butt accumulation on public premises (Table 5).

**Table 5.** Self-reported accountability for cigarette waste littering.

| Perceived Responsible Entity | Strongly Disagree | Disagree | Neither Agree nor Disagree | Agree | Strongly Agree |
|---|---|---|---|---|---|
| Smokers, because they improperly throw cigarette butts in public locations | 19.7% | 19.9% | 19.4% | 19.5% | 21.5% |
| Cigarette producers, because they produce filters that pollute the environment | 19.8% | 19.8% | 19.7% | 20.8% | 20.8% |
| Local administrations, because they should provide more places for proper disposal of cigarette butts | 19.8% | 19.7% | 20% | 18.2% | 22.3% |
| Central government, because they should take more action to discourage cigarette butt pollution | 18.9% | 20.1% | 18.9% | 20.8% | 20.8% |

## 4. Discussion

*4.1. Theoretical and Practical Implications*

Our findings are consistent with prior observational studies that found that at least 65% of the smoking population discards cigarette filters, which is a larger percentage than the rate of discarding other disposable products [23]. This is largely due to the prevalent misperception that cigarette butts are biodegradable and made of plastic. Similarly, another study concluded that the following aspects are favoring cigarette butt buildup on

Spain's southern coast: seasonality and number of beach visitors, beach typology (remote, rural, village, or urban areas), and the frequency of cleaning activities conducted [24].

As shown in Table 4 in the Results section, a significant portion of the population carelessly discards cigarette butts because they believe it to be the norm, not realizing the detrimental effects this waste has on the environment and, implicitly, on the population. These factors, along with the absence of particular facilities like trash containers and smoking areas, may be a significant contributing cause to this type of littering. Similar findings were reached by Rath and colleagues, who studied 1000 smokers from the United States and found that individuals who did not consider cigarette butts to be litter were more than three and a half times more likely to admit to having ever done so [15].

Taking into account the low level of environmental knowledge on cigarette waste and its impact, information on the costs of the careless disposal of cigarette waste must be made available to consumers along with the key benefits that can be leveraged to encourage behavior change (e.g., increased quality of life brought on by pollution reduction, financial rewards for participation in recycling programs, or new jobs resulting from implementing circular economy ideas). Also, previous studies showed that age might influence smokers' behavior towards littering, as young respondents generally litter more frequently than older smokers [11]. Although we could not identify this pattern in the current results, this might be explained by the occurrence of social desirability, as respondents tend to offer socially acceptable answers to questions concerning problematic behaviors such as littering in public spaces.

From a practical standpoint, instilling pro-environmental social norms among the smoker population might lower the level of pollution and improve the overall quality of life. Social marketers should emphasize the useful input of proper cigarette waste disposal techniques to a circular economy, as they have the potential to provide job opportunities. Social marketers must also create teaching initiatives to help people match their ecological views with their actual actions and behavior. The present environmental challenges pertaining to climate change such as heat islands, massive floodings, or droughts may be used to argue that environmental wellbeing should be prioritized [25,26].

### 4.2. Limitations and Future Research Directions

This research presents certain inherent limitations, despite the fact that it attempts to make insightful contributions to the efforts made for solving the issue of cigarette waste littering. The cross-sectional nature of the data collection used in this study restricts the findings' generalizability, which might be tackled by another follow-up quantitative study that is more typical of the population. Such a study might also concentrate on determining how social norm aspects like descriptive norms, subjective norms, and cigarette waste disposal behavior interact with one another.

An important research direction would be to determine whether there is a link between littered cigarette waste and various types of land utilization. It is possible that different types of land use (e.g., socialization, sport activities, cultural, industrial) are related to distinct littering behavioral patterns. However, more research initiatives in this area are required to establish specific associations between littered cigarette waste and land-use types [27]. Monitoring programs relying on similar findings, such as the Cigarette Butts Pollution Index, which evaluates the possible pollutants from cigarette butts leaking into the soil, could be a promising future study topic [28].

Furthermore, while controlling for the population density factor, future research should look into the smoking habits of smokers throughout the summer to see if there are any underlying variables that could explain the differences between summer and other seasons [29]. High levels of waste, particularly in coastal locations, become a concern during the summer months as a result of the growth in the population at tourism-related destinations. The amount of waste may double in the summer compared to other times of the year [25]. These increases in waste necessitate the need for beach cleanup procedures,

which are highly expensive. Therefore, all of these cleaning initiatives are merely a temporary remedy, as the real issue resides with the users, who are the root of the problem. Also, in order to tackle the littering problem in a comprehensive manner, studying the effectiveness of existing littering laws and their enforcement in Romania could provide further insights into the issue that could lead to better regulations.

## 5. Conclusions

Following the present results and those of previous study on the effects and behavior related to inadequately disposed cigarette butts, we conclude that it is critical to build a more mindful management of this waste, as well as educate citizens about the damage it does. Given the low rate of environmental knowledge identified through the developed questionnaire and the diffuse sense of accountability among smokers, producers, and authorities, we can state that the cigarette littering in Romania continues to be a persistent environmental issue despite all of the recent efforts sustained in this direction [30].

It is our belief that before anything else, consumers need to understand how their careless behavior impacts both the land and aquatic environments. The procedure for managing and processing such waste must first be developed, but the transformation as a whole must begin with consumer awareness. Even though recent years have witnessed considerable effort put into dealing with cigarette waste, a sustainable method of disposal that has a high rate of acceptance among consumers has not yet been discovered. The circular economy holds great potential in this matter; however, in order to transform the waste fraction from cigarette butts into marketable raw material, new concepts and methods must be created. Several difficulties are encountered in the process of recycling this hazardous waste, including technical or management issues, economic viability, and legislative obstacles. Moreover, public authorities dealing with severe littering rates could be persuaded to take action in this regard by performing cost–benefit analysis in collaboration with research institutes or NGOs, in order to determine the viability of cigarette butt littering prevention or reduction initiatives.

Lastly, with respect to the littering behavior, we conclude that a comprehensive behavioral framework should be developed further in order to create and assess behavioral interventions targeted at littering. The two general distinct approaches to reduce littering behavior proposed by Geller et al. [31] should be the baseline for creating interventions tailored to all pre- and post-consumption stages. According to the first approach, goal-setting, commitment, and demonstration tactics can all be used to alter the predisposing factors that lead to littering behavior [32]. Regarding the second approach, there are programs that concentrate on the negative effects of littering behavior by either rewarding abstinence or imposing fines or levies. To make progress in our quest to reduce cigarette waste, both points of view should be examined jointly [33]. Nevertheless, it is essential to stress that, despite being generally a useful instrument for studying littering behavior, the interview survey approach employed in this study has several drawbacks. Thus, in order to develop comprehensive litter prevention programs, local government entities cannot exclusively rely on survey results, but nonetheless, such surveys can yield significant findings that might be utilized in developing litter prevention tools [34–36].

**Author Contributions:** Conceptualization, E.S.L. and L.I.C.; methodology, E.S.L., A.S., L.I.C. and E.C.R.; formal analysis, E.S.L. and A.L.B.; investigation, E.S.L., A.S. and A.L.B.; resources, E.S.L.; data curation, L.I.C. and A.S.; writing—original draft preparation, E.S.L., A.S. and A.L.B.; writing— review and editing, L.I.C. and E.C.R.; supervision, E.S.L. and E.C.R.; funding acquisition, E.S.L. All authors have read and agreed to the published version of the manuscript.

**Funding:** This work was supported by a grant from the Institute for Research in Circular Economy and Environment "Ernest Lupan", project number CI499/25.10.2021.

**Institutional Review Board Statement:** The study was conducted in accordance with the Declaration of Helsinki, and approved by the Ethics Committee of Institute for Research in Circular Economy and Environment Ernest Lupan (protocol code 122/20.05.2019l, date of approval 227/04.05.2023).

**Informed Consent Statement:** Informed consent was obtained from all subjects involved in the study.

**Data Availability Statement:** Not applicable.

**Conflicts of Interest:** The authors declare no conflict of interest.

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
