# Peer review of "Smokers’ Attitude and Behavior towards Cigarette Littering in Romania: A Survey-Based Approach"

_sustainability, doi:10.3390/su151511908_

Round 1

Reviewer 1 Report

CENTRAL AND GENERAL ISSUES

Summary

En el paper de analiza el tópico de las colillas de cigarrillos, que continúan siendo un desafío significativamente perjudicial tanto para la salud humana como para la calidad del medio ambiente y la vida en general. El análisis se basó en una survey approach. De acuerdo con los hallazgos, la conciencia sobre los impactos y las características de las colillas de cigarrillos es problemática, al igual que las explicaciones de los propios fumadores sobre su conducta. I believe that the result adds value to the literature on substance abuse. However, I think there are some important aspects that need to be improved before recommending its publication in Sustainability.

Specific Comments

1. The authors apply a widely used technique. This is a widely used technique, and I don't think they should change it. However, two recent articles point out that environments are becoming more complex and require advanced analysis techniques to control smoking and drug use, such as Machine Learning or Artificial Intelligence. In this line, the authors must include a paragraph in the introduction that talks about it and, in addition, it is mandatory to cite these reference works where they talk about it:

Andueza, A., Del Arco-Osuna, M. Á., Fornés, B., González-Crespo, R., & Martín-Álvarez, J.M. (2023). Using the Statistical Machine Learning Models ARIMA and SARIMA to Measure the Impact of Covid-19 on Official Provincial Sales of Cigarettes in Spain. International Journal of Interactive Multimedia and Artificial Intelligence, 8 (1), 73-87. http://dx.doi.org/10.9781/ijimai.2023.02.010

A. Suruliandi, T. Idhaya, S. P. Raja (In Press). Drug Target Interaction Prediction Using Machine Learning Techniques – A Review. International Journal of Interactive Multimedia and Artificial Intelligence, vol. In Press, issue In Press, no. In Press, pp. 1-15. https://doi.org/10.9781/ijimai.2022.11.002

2. The authors point out as a future line of research the analysis of behavior in summer, particularly in coastal locations, given that garbage becomes a concern due to the growth of the population at destinations related to tourism. I find this point interesting and its link with the seasonal behavior of cigarette sales. I think the authors should base that claim on a paper that talks about seasonality in sales.

3. In the conclusions I do not see any paragraph showing the limitations of this work. It would be important for the limitations of this paper to be made clear.

Author Response

Dear Editor and Reviewers,

Thank you very much for your valuable comments and suggestions concerning our manuscript.  Your feedback is very helpful for revising and improving our paper, thus we have carefully studied your  comments and have made several corrections which we hope meet with approval. The main corrections in the manuscript, marked with ”track changes” function. Our response to the reviewer’s comments are as following:

Specific Comments

  • The authors apply a widely used technique. This is a widely used technique, and I don't think they should change it. However, two recent articles point out that environments are becoming more complex and require advanced analysis techniques to control smoking and drug use, such as Machine Learning or Artificial Intelligence. In this line, the authors must include a paragraph in the introduction that talks about it and, in addition, it is mandatory to cite these reference works where they talk about it:

Andueza, A., Del Arco-Osuna, M. Á., Fornés, B., González-Crespo, R., & Martín-Álvarez, J.M. (2023). Using the Statistical Machine Learning Models ARIMA and SARIMA to Measure the Impact of Covid-19 on Official Provincial Sales of Cigarettes in Spain. International Journal of Interactive Multimedia and Artificial Intelligence, 8 (1), 73-87. http://dx.doi.org/10.9781/ijimai.2023.02.010

Suruliandi, T. Idhaya, S. P. Raja (In Press). Drug Target Interaction Prediction Using Machine Learning Techniques – A Review. International Journal of Interactive Multimedia and Artificial Intelligence, vol. In Press, issue In Press, no. In Press, pp. 1-15. https://doi.org/10.9781/ijimai.2022.11.002

Thank you very much for this suggestion, this techniques are indeed very interesting and we mentioned them in the introduction section with the refferences  you kinldly provided.

  1. The authors point out as a future line of research the analysis of behavior in summer, particularly in coastal locations, given that garbage becomes a concern due to the growth of the population at destinations related to tourism. I find this point interesting and its link with the seasonal behavior of cigarette sales. I think the authors should base that claim on a paper that talks about seasonality in sales.

We took into account your suggestion and provided a reference to a paper discussing seasonality in sales concerning cigarettes.

  1. In the conclusions I do not see any paragraph showing the limitations of this work. It would be important for the limitations of this paper to be made clear.

Thank you for your remark, we decided to discuss limitations in the discussion sections in order to allign them with possible research directions, for example “This research presents certain inherent limitations, despite the fact that it attempts to make insightful contributions to the efforts made for solving the issue of cigarette waste littering. The cross-sectional nature of data collection used in this study restricts the findings' generalizability, which might be tackled by another follow-up quantitative study that is more typical of the population”.  Nonetheless, we made a refference to the most important limitation in the conclusion section.

Sincerely,

The authors

Reviewer 2 Report

Dear Authors,

The provided article is relevant and important in that it does not often refer to the study of smoker's attitude. But despite the relevance and sufficient scientific justification, the article needs to be improved.

The article provides an in-depth examination of the behaviors, attitudes, and beliefs of Romanian smokers about cigarette waste and its disposal. The approach to this topic is well organized, thorough, and grounded in sound empirical research. The survey-based methodology is robust, and the sample size of 1643 is reasonably large, leading to credible results.

The introduction effectively contextualizes the issue within the larger framework of environmental pollution and provides a comprehensive overview of the problem at hand. The authors have thoroughly researched the subject and referenced a wide array of sources, thus strengthening the article's credibility.

The methodology section provides a detailed explanation of the data collection method, including the design of the study, sample selection, and questionnaire deployment. The authors' thorough and transparent approach to describing their methodology allows for the replication of the study, which is commendable.

The results section is well presented with detailed tables and charts. It provides an overview of smoking habits, littering motivations, and environmental knowledge of the respondents. The self-reported accountability portion is insightful, showing how responsibility for cigarette litter is diffused across several entities, from the smokers themselves to the central government.

However, the article could be improved with more analysis on the data collected. While the raw results are presented in detail, there is a lack of interpretation or discussion of the implications of these results. Some of the findings could have been elaborated upon, for instance, the link between respondents’ perceived accountability for cigarette waste and their behaviors or attitudes towards littering. This would have made the paper more informative and would have helped bridge the gap between the results and discussion sections.

The discussion section highlights the theoretical and practical implications of the study, which is quite useful. It helps to provide actionable steps and suggest potential policy interventions. However, this section could be improved by including comparisons with similar studies in other countries. This would allow for a better understanding of the uniqueness or commonality of the problem in the context of Romania.

Lastly, the conclusion gives a solid summary of the study's findings and reiterates the significance of the issue at hand. However, the authors could have provided more specific recommendations based on their results. The mention of creating a behavioral framework to reduce littering is promising, but the details of how to achieve this are scant. More actionable insights would make the conclusion stronger.

Overall, the article is well-written, well-structured, and presents a compelling case for the importance of addressing cigarette littering in Romania.

 Recommendations:

 1.     In the results section, consider adding more analysis or interpretation of the data.

 2.     In the discussion section, include comparisons with similar studies in other countries.

 3.      In the conclusion, provide more specific recommendations based on the study's results.

 4.       Consider exploring the demographic and socio-economic variables that may influence littering behavior and attitudes.

5. More specific results need to be specified in the Abstract.

6. In the text it is necessary to cite according to the requirement of the journal in square brackets and reference in MDPI style.

 7.       Consider studying the effectiveness of existing littering laws and their enforcement in Romania, which could provide further insights into the issue.

The text needs English polishing

Author Response

Dear Editor and Reviewers,

Thank you very much for your valuable comments and suggestions concerning our manuscript.  Your feedback is very helpful for revising and improving our paper, thus we have carefully studied your  comments and have made several corrections which we hope meet with approval. The main corrections in the manuscript, marked with ”track changes” function. Our response to the reviewer’s comments are as following:

 Overall, the article is well-written, well-structured, and presents a compelling case for the importance of addressing cigarette littering in Romania.

 Recommendations:

  1. In the results section, consider adding more analysis or interpretation of the data.

Thank you very much for this recommendation, we added more interpretation where it was appropiate in the result section.

  1. In the discussion section, include comparisons with similar studies in other countries.

        Thank you for this suggestion, we made refferences to two other similar studies and pointed out similarities in section 4.1

  1. In the conclusion, provide more specific recommendations based on the study's results.

We developed futher the conclusion section by taking into account your suggestion and pointing out more recommendations.

  1. Consider exploring the demographic and socio-economic variables that may influence littering behavior and attitudes.

Thank you for this suggestion, we tried to better integrate discussion regarging demographics in the updated manuscript.

  1. More specific results need to be specified in the Abstract.

Thank you for pointing this out, we modified the abstract in the updated manuscript.

  1. In the text it is necessary to cite according to the requirement of the journal in square brackets and reference in MDPI style.

We revised the bibliography according to MDPI style in the updated manuscript.

  1. Consider studying the effectiveness of existing littering laws and their enforcement in Romania, which could provide further insights into the issue.

Thank you for this suggestion, this might be indeed an interesting research direction so we included it in section 4.2.

Sincerely,

The authors